

# A near-surface sea temperature time series from Trieste, north Adriatic Sea (1899-2015)

Fabio Raicich[1], Renato R. Colucci[1]

[1]CNR, Institute for Marine Sciences, Trieste, I-34149, Italy

*Correspondence to*: Fabio Raicich (fabio.raicich@ts.ismar.cnr.it)

**Abstract.** A time series of near-surface sea temperature was built from observations performed in the harbour of Trieste from 14 July 1899 to 31 December 2015. The description of the observation sites and instruments was possible thanks to historical documents. The measurements compose two data set: The first consists of analogue data obtained by means of thermometers and thermographs, one or two times per day, in the periods 1899–1923 and 1934–2008; the second consists of digital records

obtained by thermistors on hourly basis in the period 1986–2015. A quasi-homogeneous time series of daily temperatures at 2-m depth is formed from direct observations at that depth and from temperatures estimated from observations at shallower depths. From this time series a mean temperature rise rate of 1.1±0.3 °C per century was estimated. The data are available through SEANOE (doi: 10.17882/58728; Raicich and Colucci, 2019).

## 1 Introduction

The knowledge of the processes and evolution of the earth system depends on observations. As the oceans cover most of the earth's surface, marine data are particularly valuable and, therefore, are in high demand by public authorities, private enterprises, the scientific community and the civil society, in a wide range of research fields and applications. The public release of marine data and related products has been fostered in the framework of several European initiatives, as, for instance, EMODNet (www.emodnet.eu) and SeaDataNet (www.seadatanet.org) (Shepherd, 2018).

Long time series of observations represent a key element in climate studies, both based on data analysis and model reconstructions, and the sustainability of suitable observing systems is critical, as emphasized by IFSOO (2012). Although sea-temperature measurements date back to over 150 years ago (International Comprehensive Ocean-Atmosphere Data Set, Woodruff et al, 2011), time series longer than a few decades are uncommon, even in coastal areas.

Thanks to data retrieved from the archives of the Institute for Marine Sciences of CNR in Trieste, a quasi-homogeneous time

series of daily temperature at 2-m depth was formed from near-surface sea temperatures measured in Trieste harbour, at the northernmost end of the Adriatic Sea, from 1899 to 2015. Even though these data are coastal, the considerable length of the time series makes it not only useful for local studies but also a long-term indicator of climate change in the northern Mediterranean.



The observation site is located in the Gulf of Trieste, which is a bay of about $20\times25$ km$^2$ with a maximum depth of 25 m (Fig. 1). The shallowness and semi-enclosed character of the basin make near-surface temperature strongly dependent on the atmospheric forcing. As a result, a marked seasonal cycle is observed, ranging from 9 °C in February and 25 °C in July (period 1991-2003; Malačič et al., 2006); moreover, fast temperature changes often occur on synoptic time scales, due to bursts of north-easterly Bora wind, that causes coastal upwelling (e.g., Crisciani and Raicich, 2004; Crise et al., 2006) and intense air-sea heat fluxes (Stravisi and Crisciani, 1986; Picco, 1991; Supić and Orlić, 1999; Malačič and Petelin, 2001; Raicich et al., 2013). No significant influence on sea temperature is expected from the fresh water discharge in the Gulf of Trieste. The river runoff is concentrated in the northern part of the gulf, about 20 km from Trieste, where the mean annual discharge is 114 m$^3$ s$^{-1}$, i.e. over 95% of the total, while only 5 m$^3$ s$^{-1}$ discharge occurs from small streams along the Slovenian coast, at about 10 km from Trieste (Cozzi et al., 2012).

In the next section the data used in this work and their sources will be described. The methods used to derive the 2-m temperature time series and to estimate the related errors will be outlined in Sect. 3. Section 4 will include basic information on the data availability. Concluding remarks will be presented in Sect. 5.

## 2 Data description

### 2.1 Measurement sites and instruments

All the sites are located in Trieste harbour (Fig. 1) within an area of approximately 0.1 km$^2$; their geographical positions are summarized in Table 1. Molo (Pier) Sartorio (SAR) and Molo Santa Teresa (TER) face a semi-enclosed bay, while Bagno (bathing establishment) Savoia (SAV), Porto (harbour) Lido (LID) and Molo Bandiera (BAN) face the open sea. For each site, Table 1 summarizes the depths of measurement, the instruments used and the periods of observation.

The earliest regular observations of sea temperature date back to summer 1899. The first instrument used was a *Pinsel-Thermometer*, literally translated as brush-thermometer, manufactured by the H. Kappeller firm in Vienna, with scale pitch of 0.2 °C. In order to measure water temperature, its bulb was wrapped in a sort of brush which could absorb water and, after the thermal equilibrium was reached, keep it insulated from the surrounding environment while the instrument was extracted from the water and read (Attlmayr, 1883).

On 8 January 1934 a thermograph was put into operation , manufactured by the P. Wegener firm in Ballenstedt, Germany. The instrument was connected by a 5-m long capillary to a bulb placed at 2 m below the mean sea level at about 70 m from the shore, with chart (fixed on a rotating drum) speed of 45 mm d$^{-1}$ (~1.9 mm hr$^{-1}$) and non-linear vertical scale corresponding to 2.1 mm °C$^{-1}$ at 15 °C; the chart was changed weekly. Figure 2 shows the chart for 1–8 July 1935. On 19 June 1943 the observations ended because the connection between bulb and capillary was broken and the bulb was lost; attempts to obtain a replacement failed probably because of the ongoing Second World War (Geophysical Institute, 1943). Another thermograph was employed from 27 April 1948 to 1 December 1952 (Picotti, 1955). It was manufactured by the SIAP firm in Bologna, Italy, with chart speed of 40 mm d$^{-1}$ (~1.7 mm hr$^{-1}$) and linear vertical scale corresponding to 1.5 mm °C$^{-1}$.



Because the thermograph bulbs were at fixed height, their depth would change according to sea level variations. In few cases a very low sea level caused the bulb to be close to the sea surface, thus affecting the measurements. Trieste harbour is subject to a range of the astronomical tide up to 1.2 m on syzygies, and a mean seasonal sea-level range of approximately 15 cm, mainly modulated by the atmospheric forcing (pressure and dominant winds) and the steric effect. On synoptic time scales the

atmospheric forcing has produced positive sea-level surges (storm surges) up to 1.7 m above the mean sea level and negative surges up to 1 m below (Beretta et al., 2005; Tsimplis et al., 2009; Raicich, 2010).

The effect on temperature would consist of an anomalous increase due to solar heating of the surface layer, observed at low tide and in weak wind conditions, particularly during late spring and summer sunny days. We know that in the 1934–1943 period the curve was usually interpolated or smoothed manually to correct the anomalous temperature values (Polli, 1946).

Although no explicit information is available for the 1948–1952 period, the same procedure was also likely adopted.

Well thermometers have been usually employed for direct measurements, generally to fill gaps in the thermograph records. The instruments were manufactured by SIAP and their scale pitch was 0.1 °C.

In 1986 automatic temperature recording started at Molo Bandiera using PT100 thermistors, having a nominal accuracy of approximately 0.1 °C. Usually, ocean temperature should be measured to the 0.01 °C accuracy, i.e. one order of magnitude

better, nevertheless, the PT100 accuracy is considered sufficient to measure the highly variable near-surface sea temperature in Trieste harbour, which is strongly influenced by the atmospheric forcing.

We will distinguish two main data sets. Set A consists of manual and analogue automatic measurements performed from 14 July 1899 to 31 December 2008. Set B consists of digital records from 18 November 1986 to 31 December 2015. Several gaps exist due to instrumental malfunctions and failures, particularly in the earlier years of each data set, and some missing data

may have simply been lost.

## 2.2 Data set A (1899-2008)

The Maritime Observatory (Osservatorio Marittimo) of Trieste managed the earliest sea-temperature observations from 14 July 1899 to 31 December 1920. They were either reported as an additional column on tables of hourly air temperature data, obtained from the digitization of thermograph charts, or on separate sheets. The sheet for 1916 is shown in Fig. 3. The data for

1905, 1909–1914, 1918–1919 are missing. According to archive documents, the observations were also performed at least in the second half of 1918 and the first half of 1919 (Maritime Observatory, 1919), but the data could not be found. No information is available for the other data gaps.

The data sheet for July 1899 reports that the measurements were performed at the head of Molo Sartorio at 0.75-m depth at 9 AM (Central European Time). This was the situation until 1904, while in 1906–1908 the time changed to 10:30 AM. In 1915–

1917 the site was moved by approximately 300 m to Molo Santa Teresa near the lighthouse and the observation time was 8 AM. In 1920 temperature was measured at 9 AM, again at Molo Sartorio.

We can only assume that the nominal depth remained 0.75 m, as after the very first data sheet of 1899 no further information was provided. This assumption seems reasonable because the instrument was only suitable for near-surface water





measurements. In 1915-1917 the data were also published in the daily telegraphic weather bulletin (Bollettino Meteorologico Telegrafico) issued by the Maritime Observatory (Maritime Observatory, 1915-1917).

No data are available for the 1921-1933 period except few sparse measurements in 1922-1923.

In 1934 the observations were resumed by the Geophysical Institute (Istituto Geofisico), which was renamed the
Thalassographic Institute (Istituto Talassografico) in 1941. Temperature at 12 Noon was usually extracted from the thermograph records for publication. Manual measurements with a well thermometer were performed in case of thermograph failure, nevertheless several gaps still affect the record, particularly during 1941-1944.

Manual observations continued until 2008 and daily values were reported in monthly summaries alongside the meteorological data. From 1944 to 1979 they were performed at Bagno Savoia at 8 AM and 7 PM; in 1952–1957 an additional observation
was added at 12 Noon. From 1964 to 1970 temperature was also measured at Molo Sartorio in the morning (times were reported). On an unknown date between 1971 and 1975 the site was moved to Porto Lido, 180 m from Bagno Savoia (Fig. 1). From 1980 to 2008 the observation was nominally made at 10 AM.

The measurement sites can be divided in two groups. The first consists of Molo Sartorio and Molo Santa Teresa, located inside the Sacchetta Basin (Fig. 1); the second is composed of Bagno Savoia, Porto Lido and Molo Bandiera, outside the Sacchetta,
facing the open sea. Almost simultaneous (within half an hour) observations allow to compare Molo Sartorio with Bagno Savoia (1295 'surface' data in 1964–1968) and with Porto Lido (299 data at 2 m depth in 1983–1986). The average temperature differences are negligible in both cases (0.0±0.6 °C) and it is therefore possible to merge the different data records into a quasi-homogeneous time series.

Regarding the times of observation, it is certain that permanent staff was on duty until 1919 at Molo Sartorio and Molo Santa
Teresa in offices close to the observation sites, thus deviations from nominal times were probably a few minutes. By contrast, from 1934 onwards direct measurements required the observer to move from his office to the observation site; therefore, the observation time was variable and sometimes weather conditions were so bad to prevent the observations from being performed. The actual time was reported in many cases, but this was not the rule.

These analogue records were reported to the 0.1 °C precision, occasionally 0.05 °C.

**2.3 Data set B (1986-2015)**

From 1986 to 1989 a PT100 probe was used at 0.4-m depth, which can be deemed equivalent to the 'surface' depth of data set A (Table 1). In 1992 the depth was changed to 2 m and in 1994 temperature began being measured at both 0.4- and 2-m depths. In 1999 three probes were installed at 0.4-, 2- and 6-m (sea bottom) depths; this configuration was kept since. These probes were attached to a float in order to keep their depth constant.

The original data record consists of hourly temperatures, represented by 10-min averages at the end of the hour; from mid-2008 onwards average, minimum and maximum temperatures were recorded every 10 minutes. The record is affected by several interruptions due to system failures; sometimes the probes were damaged or torn away by waves during storms. The measurements were originally reported to the 0.01 °C precision.



## 3 The time series at 2-m depth

A composite long-term time series of mean daily temperatures at 2-m depth was built merging the two data sets described in in the previous section. When not available from direct measurements, the 2-m temperature was estimated from the observations at other depths.

### 3.1 The estimate of mean daily temperature

A provisional daily mean was first obtained by averaging the original observations for each calendar day, even when only one observation was available. Although the mean daily temperature range is generally within 0.5 °C, the daily means computed with few observations may significantly differ from those obtained with 24 hourly data, which can be regarded as a standard. To account for the bias, mean corrections were estimated using data set B (see Sect. 2.3).

For each depth (0.4, 2 and 6 m) and each calendar day (1 January-31 December), climatological values for the 1999–2015 period were obtained by averaging hourly (0-23) temperatures and mean daily temperatures of days when all the 24 hourly observations were available, in order to account for the complete daily cycle.

If $h$ represents the hour, $d$ the day, $m$ the month, $y$ the year, and $z$ the depth, let $T_o(h,d,m,y,z)$ be the observed temperature, $T_c(h,d,m,z)$ the climatological hourly temperature and $T_{24c}(d,m,z)$ the climatological mean daily temperature:

$$T_c(h, d, m, z) = \sum_{y=1999}^{2015}[T_o(h, d, m, y, z) \cdot p(h, d, m, y, z)]/\sum_{y=1999}^{2015} p(h, d, m, y, z), \qquad (1)$$

$$T_{24c}(d, m, z) = \sum_{h=0}^{23} T_c(h, d, m, z)/24, \qquad (2)$$

$p$ being a weighing factor, equal to 1 if $T_o$ is available and 0 if it is not. For a given $(h,d,m,z)$, in 1999–2015 there are generally between 13 and 17 values of $T_o$. A 31-day running mean is subsequently applied to $T_c$ and $T_{24c}$ in order to smooth out the effect of outliers. The values for 0.75-m depth are obtained by linear interpolation between those of 0.4- and 2-m depths, and the values for 29 February are interpolated using those of 28 February and 1 March.

The mean daily temperature $T(d,m,y,z)$ is estimated as:

$$T(d, m, y, z) = T_{24c}(d, m, z) + \frac{1}{N}\sum_{k=1}^{N}[T_o(t_k, d, m, y, z) - T_c(t_k, d, m, z)], \qquad (3)$$

Where $N=N(d,m,y,z)$ is the number of available observations on the relevant day and $t_k$ is the time of observation; when $T_c$ is not a full hour, it is interpolated at the proper time using the nearest hourly values. The term in square brackets of Eq. 3 represents the departure of the observed value from climatology. Note that, if $T_o$ is available at full hour from 0 to 23, then $T$ is the arithmetic average of the 24 observations.

In other words, the difference between an observation and the corresponding climatological value provides a constant that is used to re-scale the climatological daily cycle; then, the average of the re-scaled daily cycle represents the estimated daily temperature. If more observations are available in a given day, the final estimate is the average of the individual estimated values.



When the 2-m observations are unavailable, the daily means are possibly estimated from data at other depths. Let $Z=2\ m$, then:

$$T(d, m, y, Z) = T(d, m, y, z) - [T_{24c}(d, m, z) - T_{24c}(d, m, Z)], \qquad (4)$$

therefore:

$$T(d, m, y, Z) = T_{24c}(d, m, Z) + \frac{1}{N}\sum_{k=1}^{N}[T_o(t_k, d, m, y, z) - T_c(t_k, d, m, z)], \qquad (5)$$

The term in square brackets on the right-hand side of Eq. 4 represents the correction that accounts for the incomplete representation of the daily cycle. Considering that, overall, the observations were made in the 8 AM–7 PM interval (sect. 2.2, above), the largest corrections, in absolute value, occur in June and July when values between -0.3 and 0.4 °C are expected, depending on the time of observation, with an average of +0.1 °C; the smallest corrections are found between mid-October and mid-February with values between -0.1 and +0.1 °C and average around 0.0 °C. These corrections concern estimates from

single observations, as in data set A, while they are negligible in data set B where the complete daily cycle is generally available.

The term in square brackets on the right-hand side of Eq. 5 represents the correction related to the daily temperature estimate from data at depths other that 2 m. In this case, the corrections range between 0.02 °C in late November-early December and 0.68 °C in late June-early July. Larger corrections are a consequence of the water column stratification which typically occurs

in summer.

### 3.2 The error on mean daily temperature

The error on $T$, namely $\sigma$, is computed from those on the observation, $\sigma_o$, and on the climatologies, $\sigma_{24c}$ and $\sigma_c$, respectively. They are assessed semi-empirically as follows.

An observation is basically affected by an instrumental error and an environmental error. The error on temperature measured

with manual thermometers can be estimated in half the scale pitch, namely 0.05 or 0.1 °C, depending on the instrument. In the case of thermographs, the distance of 1.5-2 mm between the chart markings and the curve thickness probably determine a reading error of approximately 0.2 °C. The environmental error is caused by small temperature fluctuations occurring in the water body at sub-hourly frequencies, due to turbulence and circulation, therefore an instantaneous measurement may not be really representative of the hourly value. In order to estimate this error, average hourly temperature ranges are computed using

the temperature extremes that are available for 2008–2015. The mean 1-hr temperature ranges turn out to be 0.24, 0.30 and 0.29 °C at 0.4, 2 and 6 m, respectively. A ±0.15 °C error (0.3 °C band) can be adopted for all depths.

An additional error often affects the observations of data set A because not the actual observation times but only nominal times are reported. This time uncertainty reflects on temperature because of the involvement of $T_c$, which is a function of $h$, in Eq. 1. As discussed in Sect. 2.2, the time uncertainty until 1917 can be considered negligible since the observation site and the

observers' office were nearby. An estimate of the time uncertainty on manual observations from 1920 onwards can be made from the known observation times in 1964-1968 and 1983-1986, obtaining approximately ±2 hours. Temperature data can be





reliably obtained from the thermograph curves every half an hour, therefore the time error can be estimated in about ±15 minutes. The time-related error is estimated in the same way used for the environmental error, in this case by analysing 4-hr and 0.5-hr intervals. The mean 4-hr temperature ranges are 0.50, 0.54 and 0.52 °C and the mean 0.5-hr ranges are 0.17, 0.23 and 0.21 °C, at 0.4, 2 and 6 m, respectively. Errors of ±0.25 °C (0.5 °C band) and ±0.1 °C (0.2 °C band) can be adopted in

these cases. Time uncertainty does not affect data set B significantly, because time is correct within a few minutes (at worst) depending on clock stability.

Table 2 summarizes the errors that can be expected in the different cases, as a function of $\sigma_T$, which is related to a known or nominal observation time, and the number of daily data available. Provided that $T_c$ and $T_{24c}$ are computed from Eq. 1 and 2, from the observational error $\sigma_o=0.18$ °C we obtain $\sigma_c=0.05$ °C and $\sigma_{24c}=0.01$ °C. Finally, the overall error on the daily

temperature estimate (Eq. 3) can be written:

$$\sigma(d,m,y,Z) = \left[\sigma_{24c}^2 + \frac{1}{N^2}\sum_{k=1}^{N}(\sigma_o^2 + \sigma_c^2)\right]^{\frac{1}{2}}, \tag{4}$$

Again, $N=N(d,m,y,z)$ is the number of available observations. To the precision of 0.1 °C, the daily means at 2-m depth are characterized by overall errors between 0.2 and 0.4 °C for data set A and between 0.1 and 0.2 °C for data set B.

**3.3 The composite time series**

Before merging data sets A and B a consistency check was done by comparing each other in the 1986–2008 period, when temperature was measured both at Porto Lido (LID) and Molo Bandiera (BAN) (Fig. 1). We computed mean temperature differences ($\Delta T = T_A - T_B$) when simultaneous measurements were available. Two periods can be distinguished, namely 18 November 1986–30 November 1989, in which $\Delta T = 0.5\pm0.5$ °C, and 1 March 1993–31 December 2008, with $\Delta T = -0.1\pm0.4$ °C, respectively. As the procedures and instruments used for direct measurements (data set A) have not significantly changed

before and after 1986, the temperature difference in the first period is likely related to the early digital records (data set B), perhaps due to instrumental drifts or inaccurate calibration. In the second period ΔT is negligible.

The composite time series was built using data set A as a basis and data set B to fill gaps and extend the time series after 2008. For a few gaps in 1987–1989 the temperature from data set B was increased by 0.5 °C, as discussed above.

**4 Data availability**

Figure 4 illustrates the data availability on a monthly basis as percentage of valid days. Mean annual temperatures were computed for complete years, i.e. having 12 valid months; a valid month has at least 80% valid days (Fig. 5). The threshold was chosen arbitrarily. Figure 4 illustrates the data availability on a monthly basis as percentage of valid days.

The 1899–2015 composite time series is characterized by fluctuations on decadal time scales and a linear trend of 1.1±0.3 °C per century (at the 95% confidence level). Because of the many missing years in the first part, this result should be taken with caution. The trend of the continuous 1946–2015 period is 1.3±0.5 °C per century.

The data used in this work are available from SEANOE as "Trieste 1899-2015 near-surface sea temperature", which includes the composite time series, data sets A and B and the monthly and annual mean temperatures (doi: 10.17882/58728; Raicich and Colucci, 2019).

## 5 Conclusive remarks

Near-surface sea temperatures data measured at different sites in the harbour of Trieste were collected, forming two data sets. The first consists of analogue data from 1899 to 2008, the second consists of digital records from 1986 to 2015. Their merger allowed to build a 117-year-long quasi-homogeneous time series of mean daily temperature at 2-m depth. Although the data accuracy is lower than the modern standards for ocean temperature measurements, these data sets represent a valuable tool to study sea-temperature variability from the synoptic time scale, connected to the meteorological forcing, to decadal and secular time scales, related to climate changes.

The search for undiscovered data will continue to possibly fill the existing gaps.

## Author contribution

FR retrieved the archived data, prepared the data sets and lead the writing of the paper. RRC was involved in temperature measurement and processing, and collaborated to the paper writing.

## Competing interests

The authors declare that they have no conflict of interests.

## Acknowledgements

The authors acknowledge the work done during over a century by the previous staff of the Maritime Observatory, the Geophysical Institute, the Thalassographic Institute, and the Institute for Marine Sciences, who managed and performed the observations, and processed and preserved the data used to build the data set. In particular, the authors would like to thank M. Iorio and E. Caterini, in the current staff of the Institute for Marine Sciences of CNR.





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




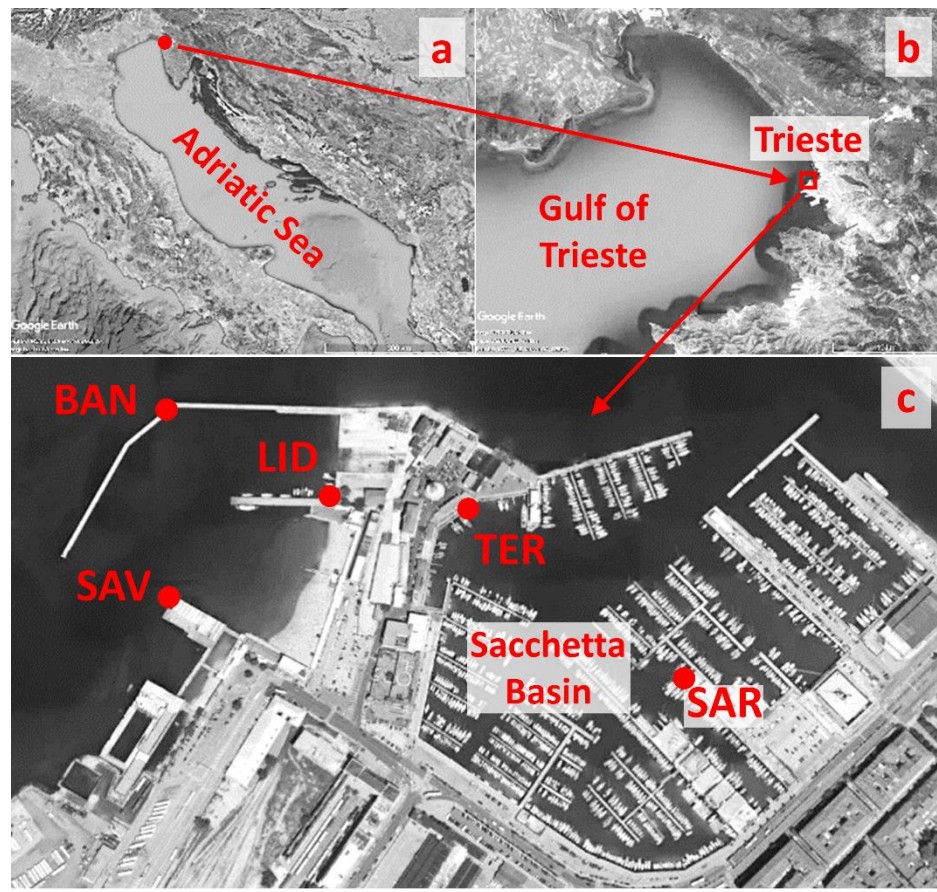

**Figure 1: Aerial image of the area of Trieste harbour with the observation sites: Molo Sartorio (SAR), Molo Santa Teresa (TER), Bagno Savoia (SAV), Porto Lido (LID) and Molo Bandiera (BAN). (Images extracted from Google Earth; © 2018 Landsat/Copernicus, © 2018 CNES/Airbus, © 2018 Digital Globe, © 2018 TerraMetric.)**





**Figure 2: Sea temperature summary sheet for 1916. On top the indication of site and observation time: "Trieste (Lighthouse)", "Sea temperature in °C observed at 8h".**

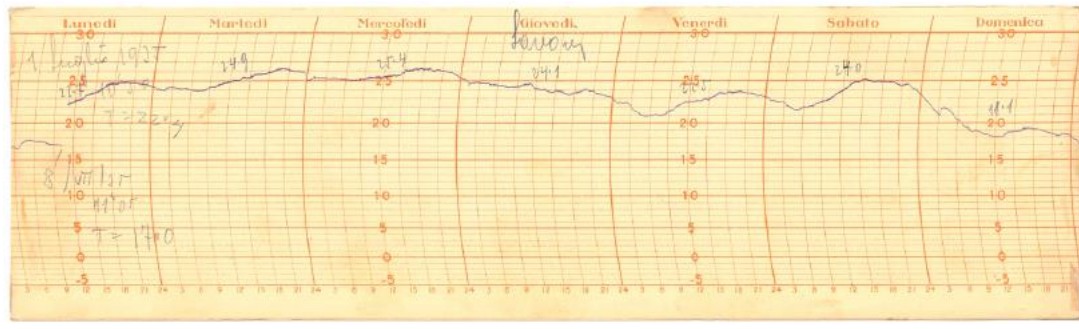

**Figure 3: Weekly thermograph chart from Bagno Savoia for 1–8 July 1935. Values written above the curve are temperatures at 12 Noon. A remarkable cooling from 25 to 18 °C, due to Bora-induced upwelling, is evident on 6–7 July.**



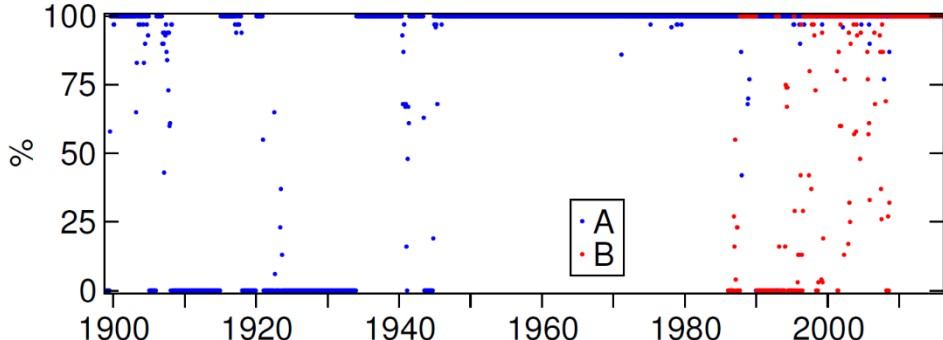

**Figure 4: Data availability on a monthly basis for data set A (1899–2008) and data set B (1986–2015).**

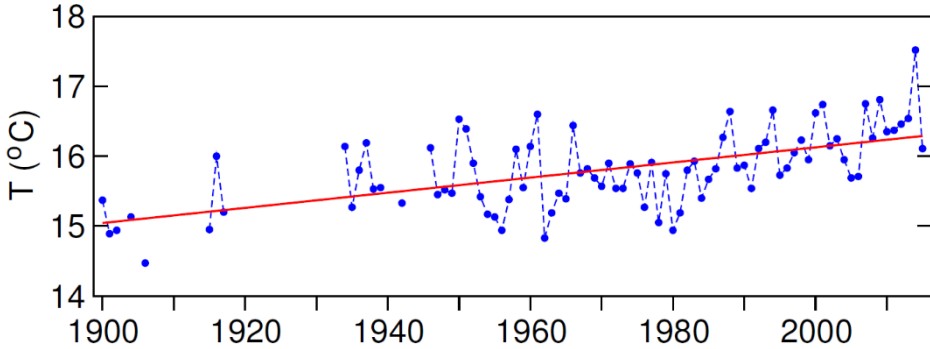

**Figure 5: Annual mean sea temperature at 2-m depth from 1900 to 2015 (blue dots and dashed curve) and linear trend (red line).**



**Table 1: Summary of observation sites, their geographical position and instruments.**

| Site | Lat N (°) | Long E (°) | Depths | Instruments | Time interval |
|---|---|---|---|---|---|
| SAR – Molo Sartorio | 45.6473 | 13.7595 | 0.75 m | 'Pinsel-Thermometer' | 1899–1908 |
| | | | 'surface' | well thermometer | 1964–1968 |
| TER – Molo Santa Teresa | 45.6489 | 13.7563 | 0.75 m | 'Pinsel-Thermometer' | 1915–1920 |
| SAV – Bagno Savoia | 45.6481 | 13.7527 | 'surface' | well thermometer | 1964–1970 |
| | | | 2 m below MSL | thermograph | 1934–1943 |
| | | | 2 m | well thermometer | 1934–1964 |
| | | | 2 m below MSL | thermograph | 1948–1952 |
| | | | 2 m | well thermometer | 1966–(1971–75) |
| | | | 5 m | well thermometer | 1965–1966 |
| LID – Porto Lido | 45.6491 | 13.7546 | 2 m | well thermometer | (1971–75)–1990 |
| BAN – Molo Bandiera | 45.6500 | 13.7522 | 0.4, 2, 6 m | thermistor | 1986–2015 |

5  **Table 2: Errors on estimated mean daily temperature. See text for the explanation of different options for time-related error and number of observation. ($\sigma_I$: instrumental accuracy, $\sigma_E$: environmental error, $\sigma_T$: time-related error, $\sigma_o$: observational error, n. obs.: number of observations per day, $\sigma$: error on estimated daily mean.)**

| Instrument | $\sigma_I$ (°C) | $\sigma_E$ (°C) | $\sigma_T$ (°C) | $\sigma_o$ (°C) | n. obs. | $\sigma$ (°C) |
|---|---|---|---|---|---|---|
| *Pinsel-Thermometer* | 0.1 | 0.15 | 0; 0.25 | 0.18; 0.31 | 1; 1 | 0.18; 0.31 |
| Thermograph | 0.2 | 0.15 | 0.1 | 0.27 | 1 | 0.40 |
| Well thermometer | 0.05 | 0.15 | 0; 0.25 | 0.16; 0.30 | 1; 1-3 | 0.16; 0.30-0.22 |
| Thermistor | 0.1 | 0.15 | 0 | 0.18 | 1-24 | 0.18-0.08 |