# Peer review of "A near-surface sea temperature time series from Trieste, north Adriatic Sea (1899-2015)"

_Earth System Science Data, 2019_

## Referee Comment (RC1) · Anonymous Referee #1 · 22 Mar 2019

Please see supplement PDF for referee comments, which may be easier to read than the below.

GENERAL COMMENTS Recommendation: Minor revisions

This manuscript gathers, develops, and describes a long-term near-surface sea temperature time series from Trieste, north Adriatic Sea in northern Italy. It uniquely uses and gathers historical documents and data records to produce a composite time series of near-surface sea temperature over a 117-year period (1899-2015). The authors discuss the differences in each of the data sets used (vertical position in the water column, horizontal location along the harbor, temporal resolution of the data sampling,

time of sampling, etc.), duly noting the caveats associated with homogenizing the data sets into a composite time series. The authors also calculate and discuss data errors and availability. The referee does think that the authors use as sound of a methodology to produce the composite time series as possible, given the limitations in data heterogeneity. The major comment that I have is that it is very difficult to follow in Table 1 the description of Data set A and B (specifically the dates of the data). Otherwise, I have several minor edits and individual scientific questions. Because most of my comments are minor in nature, I recommend overall minor revisions.

SPECIFIC COMMENTS

–Page 1 –Line 12: "1.1+/-0.3 C per century was estimated": Could you mention here also the 1946-2015 1.3+/-0.5C per century trend, as that is a more continuous and thus robust trend? –Line 26: "Even though these data are coastal": What is meant by this? That coastal data may not be most useful for climate change indicators? Why? Please explain in manuscript. –Page 3 –Lines 1-10: How were the Pinsel thermometer measurements taken? Were they affected by sea level change similar to how the thermographs were affected? –Line 14: "Usually, ocean temperature should be measured to the 0.01 C accuracy": can you provide a reference for this? Seems to me that this accuracy would change based on the end user. –Lines 28-31: Are the time (9 am to 10:30 am to 8 am to 9 am) and space (300 m apart) changes too large to produce a sufficiently homogeneous dataset here? Please comment in the manuscript. –Page 4: –Lines 1-12: This is very difficult to follow along with Table 1. Suggest somehow adding "Data set A" and "Data set B" to Table 1 to make it clearer what is indicated by what in Table 1. –Line 8: "Manual observations continued until 2008" where is 2008 in Table 1? I only see dates after 1990 on the last row of Table 1, and nowhere else. –Line 10: "From 1964 to 1970": isn't it 1968 in Table 1 row 2? –Line 12: "From 1980 to 2008" where is 2008 in Table 1 besides at the bottom? –Line 16: Where is 1983-1986 in Molo Sartorio in Table 1? –Line 19: Where is 1919 in Molo Sartorio in Table 1? –Line 25: "2.3 Data set B (1986-2015)": Is this all in just the last row of Table 1? If so, how is

Data set A going to 2008? (can't find this in Table 1). –Line 27: "In 1992" What about 1989-1992? –Page 5 –Lines 3-4: How was 2-m temperature estimated from the observations at other depths? Please briefly explain in manuscript. –Line 7: "Although the mean daily temperature range is generally within 0.5 C" What does this mean? How is "mean daily temperature range generally within 0.5 C"? –Line 8: "24 hourly data" what is this? Hourly data (24 per day)? Please clarify in text. Also, what does "which can be regarded as a standard" mean? Awkwardly worded. –Page 7: –Line 4: Is rounding occurring here to get +/- 0.25 C (0.5 C band) and +/- 0.1 C (0.2 C band)? What about using the maximum range, e.g. 0.54 and 0.23 C bands? –Line 9: Where is 0.18 C in Table 2? Please clarify as there are several within the 4th column. –Line 13: Where are the overall errors between 0.2 and 0.4 C for data set A and 0.1 and 0.2 C for data set B in Table 2? –Line 15: "1986-2008" again, where is this for data set A in Table 1?

TECHNICAL CORRECTIONS

Abstract

–Line 8: "The measurements compose two data set": suggest changing to "The measurements are comprised of two data sets"

Short Summary

-"We described" should be "We describe" -"variability on different time scale" should be "variability on different time scales"

1. Introduction

–Page 1 –Line 15: should be "Knowledge of the processes and evolution of the Earth system..." –Line 16: "earth's" should be "Earth's" –Line 17: "the scientific community, and civil society" –Lines 20-21: "a key element in climate studies, based on both data analysis and model reconstructions, and thus the sustainability of suitable observing systems is critical" –Page 2 –Line 3: "February to 25..." (not and)

2. Data description

–Line 28: "Figure 2 shows the chart for 1-8 July 1935" should this be "Figure 3"? If so, will likely want to change order of figures so they are referenced in order. –Line 32: "linear vertical scale corresponding to 1.5 mm C-1" is this at 15 C like the first thermograph which had a vertical scale of 2.1 mm C-1 at 15 C? If so, please indicate. –Page 3 –Line 3: "to a range of astronomical tide" –Line 15: "better; nevertheless" (change comma to semicolon) –Line 16: "which is strongly influenced by atmospheric forcing" (delete "the") –Line 21: "2.2 Data set A (1899-2008)" where is 2008 in Table 1? –Line 24: "The sheet for 1916 is shown in Fig. 2" (not Fig. 3—see Line 28 comment above, may want to switch Figures 2 and 3) –Line 25: "1909-1914, and 1918-1919 are missing" –Page 4 –Line 7: "failure;" change comma to semicolon –Line 16: What are 1295 and 299 indicating? –Line 23: "but this was not always the case" –Line 24: "These analogue records were reported to 0.1 C precision, and occasionally to 0.05 C" –Line 33: "reported to 0.01 C precision." (remove "the")

3. The time series at 2-m depth

–Page 6 –Line 1: "the daily means are possibly estimated" "possibly" is awkwardly worded, please change –Line 9: "These corrections concern estimates" "concern" is awkward, perhaps use "affect"? –Line 17: "computed from those on the observation" is awkwardly worded, please change. Also, why do you need "respectively" here? –Line 19: "An observation is affected by" (delete "basically") –Line 21: "the curve thickness determine a reading error" (delete "probably") –Line 23: "due to turbulence and circulation. Therefore. . ." –Line 24: "representative of the hourly value" (delete "really") –Line 27: change to "only nominal times, and not the actual observation times, are reported." –Line 28: "This time uncertainty affects temperature. . ." –Line 29: "the time uncertainty until 1917" should this be 1919 or 1934 based on section 2.2? –Page 7 –Line 1: "half an hour; therefore" (change comma to semicolon) –Line 15: "by comparing the data sets" –Line 23: "as discussed in the above paragraph"?

4. Data availability
–Line 25: "as a percentage of valid days" also, valid is awkward, maybe use "total days"? –Line 26: what is meant by 80% valid days here? Same question for "valid months"? –Line 27: better word for arbitrarily? Seems very unofficial. Also, remove the repeat "Figure 4 illustrates..." sentence. –Page 8 –Line 2: "in the first part of the period"?

5. Conclusive remarks

–Line 8: "Near-surface sea temperature data" (remove the "s" in temperatures) –Line 9: "from 1899 to 2008, while the second consists..." –Lines 12-13: "study sea-temperature variability on the synoptic time scale connected to meteorological forcing, and on decadal and secular time scales related to climate changes." Also, what is meant by "secular time scales"? –Line 14: "The search for undiscovered data will continue, in order to possibly fill the existing gaps."

Author contribution

–Line 16: "the data sets and led" –Line 17: "and collaborated on the paper writing"

Figures

Figures 2 and 3: these are somewhat difficult to read, may want to ensure very high quality images of these are included so readers can clearly read them.

Figure 5: Can you add the 1946-2015 trend line on this figure? Also, should the caption read "from 1899 to 2015"?

Tables

Table 1: several comments as indicated above

Table 2: –Line 5: "number of observations". "instrumental error" –What are the 1st and 2nd numbers before and after the semicolons, e.g. 0.18; 0.31? Finally, can you somehow indicate that the errors are $\frac{1}{2}$ the band (e.g. +/- 0.15 C)?

Please also note the supplement to this comment:
https://www.earth-syst-sci-data-discuss.net/essd-2019-15/essd-2019-15-RC1-supplement.pdf

---

## Referee Comment (RC2) · Vlado Malacic (Referee) · 24 Apr 2019

**Review of the manuscript:**
**A near-surface sea temperature time series fromTrieste, north Adriatic Sea (1899-2015)**
**No: essd-2019-15**

**General view**

The manuscript is clearly written,  the content is interesting, suitable for the journal ESSD. It describes a synthesis of a collection of coastal sea temperature data over 116 years in the northernmost part of the Mediterranean Sea (Trieste, Italy). This is rare and it deserves attention. A minor revision is suggested.

It seems that the paper could gain in relevance if two points would be added.
I.
There is no comparison with the trend of (surface) sea temperatures of either coastal, nor 'global' ocean sea surface temperature data. This 'global warming trend' is a hot topic, relevant nowadays. Authors confined themselves mostly to the methodology of 'combining' the data of different measurements techniques, of different sea temperature 'sampling', on elaborating the time series (filtering the data) and on the trend of sea temperature rise that they reveal from those data. There are certainly many research papers that describe centennially temperature trends elsewhere. Moreover, there are reports of IPCC (although quality reports are lately blurred with reports of IPCC meetings…) that still somehow 'matter', e.g. the IPCC Report 'Global warming of 1.5°C', in Chapter 1:  [Allen, M.R., et al., 2018: *Framing and Context. In: Global Warming of 1.5°C. An IPCC Special Report on the impacts of global warming of 1.5°C above pre-industrial levels and related global greenhouse gas emission pathways, in the context of strengthening the global response to the threat of climate change, sustainable development, and efforts to eradicate poverty* [Masson-Delmotte, et al. (eds.)]. In Press.]. There one may find a few 'useful sentences' already at the beginning, e.g.: 'Human-induced warming reached approximately 1°C (likely between 0.8°C and 1.2°C) above pre-industrial levels in 2017, increasing at 0.2°C (likely between 0.1°C and 0.3°C) per decade (high confidence)', and also 'Accordingly, warming from preindustrial levels to the decade 2006–2015 is assessed to be 0.87°C

(likely between 0.75°C and 0.99°C).' These sentences are just very modest examples about how the result (the temperature trend in the 'intestines' of the central middle Europe, facing the sea) of authors makes sense and is 'in line' with the trends others have found. There are also differences (e.g. in the trend within last 30 years) with other findings, which would well be described in Discussion. In the Introduction, though, the relevance of this particular, long time series has to be emphasized and compared with other very long term studies.

II. The second topic for which it seems just to be linked to the paper, is the matter of the sea-level rise. A brief look on publications of authors clearly shows that at least one of them has a solid reputation in 'knowing this matter well'. Authors may relatively easily combine their sea temperature rise finding with the sea level rise simply due to steric effect – they can estimate it and may also estimate the error of the estimation (they showed how nicely they know how to estimate errors…) of sea level rise due to temperature expansion of water (e.g. the effect of salinity (variability)). There is quite a large number of papers over the Adriatic and the Mediterranean Sea that handle separately the sea level rise and the temperature rise, but only a few link these two trends. This is a good chance 'to do it right'!

**Specific comments**

Page, 1. Line 16: is the text in this line in 'bold'?

Page 3, line 24: Fig. 3 is referred. Should it be the Fig. 2? There was no Fig. 2 before in the text and it looks from Figure and figure caption of Fig. 2 that this should be Fig. 2.

Page 5, line 13: $T_0(h,d,m,y,z)$ → $T_0(h,d,m,y,z)$

Page 5, line 18: '…between 13 and 17 values of $T_0$.' Could it be added 'out of (?) 24 × 365.25' on average per year?

Page 5, expression (3): In the expression (2) $T_{24c}$ is written down. However, it somehow follows from the expression (3) and the comment below it that $T_{24c}$ should be expressed as the average of $N$ values od $T_c$ the number of available

observations on the relevant day, and not the average of '24' values (expression (2)). Correct?

Page 6, expression (5): It looks OK…

Page 7, line 9: 'observational error $\sigma_0$=0.18 °C, we obtain $\sigma_c$=0.05 °C and $\sigma_{24c}$=0.01 °C' → observational error $\sigma_0$=0.18 °C, we obtain $\sigma_c$=0.05 °C and $\sigma_{24c}$=0.01 °C.

Page 7, line 23: '…was increased by 0.5 °C, as discussed above.'. Do authors refer to the line 18 in which $\Delta T = 0.5 \pm 0.5$ °C is written? If so, then they could write this more clearly and on line 18: $\Delta T = 0.5 \pm 0.5$ °C → $\Delta T = 0.5 \pm 0.5$ °C. The same for another $\Delta T$ in the same line.

Page 7, line 27: there is a redundant copy of the sentence about Figure 4 from the line 25….

---

## Author Comment (AC1) · 8 May 2019

Pages and lines of the edited manuscript, unless otherwise stated (To make reading easier, the pdf version of this document and of the edited article, with revisions, are attached as supplement1.zip.)

SPECIFIC COMMENTS

–Page 1

–Line 12: "1.1+/-0.3 C per century was estimated": Could you mention here also the 1946-2015 1.3+/-0.5C per century trend, as that is a more continuous and thus robust

trend? Answer: The relevant sentence was completed as follows: "... was estimated, while in 1946-2015 it is 1.3±0.5 °C per century." (Page 1, lines 12-13)

–Line 26: "Even though these data are coastal": What is meant by this? That coastal data may not be most useful for climate change indicators? Why? Please explain in manuscript. A: The sentence was completed as follows: "... coastal and, therefore, sensitive to local natural and anthropogenic processes, the considerable ...". (Page 1, lines 26-27)

–Page 3

–Lines 1-10: How were the Pinsel thermometer measurements taken? Were they affected by sea level change similar to how the thermographs were affected? A: The text was modified in two places, as follows: 1) Page 2, lines 24-26: "In order to measure temperature, the instrument was manually deployed to the prescribed depth and kept there until thermal equilibrium was reached. Its bulb was wrapped in a sort of brush which could absorb water and keep it insulated from the surrounding environment while the thermometer was being extracted from water and read.". 2) Page 3, lines 3-4: "While manual measurements were performed at prescribed depths below the sea surface, the thermograph bulbs were at fixed heights, therefore their depths would change according to sea level variations."

–Line 14: "Usually, ocean temperature should be measured to the 0.01 C accuracy": can you provide a reference for this? Seems to me that this accuracy would change based on the end user. A: We agree that the concept was too strong. The sentence was modified as: "Usually, ocean temperature is reported with the precision of 0.01 °C, i.e. ..." (Page 3, line 17)

–Lines 28-31: Are the time (9 am to 10:30 am to 8 am to 9 am) and space (300 m apart) changes too large to produce a sufficiently homogeneous dataset here? Please comment in the manuscript. A: As several changes occurred in observation times, depths, sites and sampling frequencies, we could not assume the time series to be

homogeneous. The only way to say if time and space changes introduced significant biases or not, was to estimate them, and this is what we did. The reviewer's question, which we believe concerns the time and space changes in general, is answered in three places: 1) At page 4, line 17 (original manuscript), where differences are shown to be negligible between sites inside and outside the Sacchetta basin. 2) At page 6, lines 6-12 (original manuscript), where the influence of time sampling turns out to be small but not negligible in summer (relative to a daily mean obtained from 24 observations per day). 3) At page 7, line 18 (original manuscript), where differences are shown to be negligible also between the manual measurements at Porto Lido and the digital records at Molo Bandiera. In order to assess the possible effect of the biases, we also computed the long-term trends of the time series without any normalization. The text in section 4 was rewritten as: "The effect of the biases induced by poor time samplings, discussed in Sect. 3.1, can be estimated by analysing the time series of the original observations; in this case, the trend is $1.0\pm0.5\,°C$ per century. Because of the many missing years in the first part of the time series, this result should be taken with caution. The trend over the continuous 1946–2015 period is $1.3\pm0.5\,°C$ per century, both using the normalized and the original data." (Page 8, lines 14-17)

–Page 4:

–Lines 1-12: This is very difficult to follow along with Table 1. Suggest somehow adding "Data set A" and "Data set B" to Table 1 to make it clearer what is indicated by what in Table 1. A: Unfortunately, Table 1 was affected by errors and we are sorry for that. The table was rewritten accepting the reviewer's remarks. (Page 15)

–Line 8: "Manual observations continued until 2008" where is 2008 in Table 1? I only see dates after 1990 on the last row of Table 1, and nowhere else. A: Corrected. (Page 15)

–Line 10: "From 1964 to 1970": isn't it 1968 in Table 1 row 2? A: Corrected. (Page 4, line 14)

–Line 12: "From 1980 to 2008" where is 2008 in Table 1 besides at the bottom? A: The line starting with "LID-Porto Lido" was corrected. (Page 15)

–Line 16: Where is 1983-1986 in Molo Sartorio in Table 1? A: This information was actually missing; it was inserted before the line starting with "TER-Molo S. Teresa". (Page 15)

–Line 19: Where is 1919 in Molo Sartorio in Table 1? A: In order to clarify the point, the text was modified as follows: "... duty in offices close to the observation sites at Molo Sartorio (in 1899-1908) and at Molo Santa Teresa (in 1915-1917 and, probably, in 1920), thus ...". (Page 4, lines 24-25)

–Line 25: "2.3 Data set B (1986-2015)": Is this all in just the last row of Table 1? If so, how is Data set A going to 2008? (can't find this in Table 1). A: The line starting with "LID-Porto Lido" was corrected. (Page 15)

–Line 27: "In 1992" What about 1989-1992? A: The text was modified to clarify that there was an interruption of the observations: "Observations were interrupted until 1992, when the depth was changed ..." (Page 4, line 32)

–Page 5

–Lines 3-4: How was 2-m temperature estimated from the observations at other depths? Please briefly explain in manuscript.

A: Sect. 3.1 describes the method. In particular, eq. 4 and 5 show how 2-m temperature was estimated. The sentence was completed as: "... other depths, as explained in Sect. 3.1.". (Page 5, line 10)

–Line 7: "Although the mean daily temperature range is generally within 0.5 C" What does this mean? How is "mean daily temperature range generally within 0.5 C"? and –Line 8: "24 hourly data" what is this? Hourly data (24 per day)? Please clarify in text. Also, what does "which can be regarded as a standard" mean? Awkwardly worded. A: The text was unclear and the whole sentence was modified as: "Although the mean

daily temperature range is generally less than 0.5 °C, a daily mean computed with few observations may be significantly differ from a mean accounting for the full daily cycle, which is well represented by 24 hourly observations." (Page 5, line 13-16)

–Page 7:

–Line 4: Is rounding occurring here to get +/- 0.25 C (0.5 C band) and +/- 0.1 C (0.2 C band)? What about using the maximum range, e.g. 0.54 and 0.23 C bands? A: We can only provide representative errors and we believe that precisions of 0.01 °C could suggest an unrealistically high accuracy. For clarity, we changed the start of the sentence at line 4 as: "Representative errors of . . .". (Page 7, line 15)

–Line 9: Where is 0.18 C in Table 2? Please clarify as there are several within the 4th column. and –Line 13: Where are the overall errors between 0.2 and 0.4 C for data set A and 0.1 and 0.2 C for data set B in Table 2? A: The table was redesigned and, to clarify both points, its caption was rewritten as: "Table 2: Errors on estimated mean daily temperature. $\sigma$I is the instrumental error, $\sigma$E is the environmental error, $\sigma$T is the time-related error, $\sigma$o is the observational error, n. obs. is the number of observations per day, $\sigma$ is the error on the estimated daily mean. Labels a) and b) correspond to zero and non-zero uncertainties on time, respectively. When n.obs. is characterized by a range, the corresponding $\sigma$ range is shown. See Sect. 3.2 for detailed explanations." (Pages 15-16)

–Line 15: "1986-2008" again, where is this for data set A in Table 1? A: As explained in the answer to the remarks to page 4, lines 12 and 25, this was corrected. (Page 15)

TECHNICAL CORRECTIONS

Abstract –Line 8: "The measurements compose two data set": suggest changing to "The measurements are comprised of two data sets" A: We corrected using "consist of" instead of the suggested "are comprised of". (Page 1, line 8)

Short Summary -"We described" should be "We describe" and -"variability on different

time scale" should be "variability on different time scales" A: Both corrections will have to be made online.

1. Introduction –Page 1

–Line 15: should be "Knowledge of the processes and evolution of the Earth system…" A: Corrected. (Page 1, line 15)

–Line 16: "earth's" should be "Earth's" A: Corrected. (Page 1, line 16)

–Line 17: "the scientific community, and civil society" A: Corrected. (Page 1, line 17)

–Lines 20-21: "a key element in climate studies, based on both data analysis and model reconstructions, and thus the sustainability of suitable observing systems is critical" A: Corrected. (Page 1, lines 20-21)

–Page 2 –Line 3: "February to 25…" (not and) A: Corrected. (Page 2, line 3)

2. Data description –Line 28: "Figure 2 shows the chart for 1-8 July 1935" should this be "Figure 3"? If so, will likely want to change order of figures so they are referenced in order. A: We are sorry for the mistake. The text was correct, while Figures 2 and 3 at the end of the manuscript were swapped and identified by wrong numbers. The mistake was corrected. (Pages 12-13)

–Line 32: "linear vertical scale corresponding to 1.5 mm C-1" is this at 15 C like the first thermograph which had a vertical scale of 2.1 mm C-1 at 15 C? If so, please indicate. A: The indication "at 15 °C" was required for the first instrument which has a non-linear scale (line 28), while it is redundant for the second instrument, having a linear scale. The text was not modified.

–Page 3

–Line 3: "to a range of astronomical tide" A: Corrected. (Page 3, line 5)

–Line 15: "better; nevertheless" (change comma to semicolon) A: Corrected. (Page 3,

line 18)

–Line 16: "which is strongly influenced by atmospheric forcing" (delete "the") A: Corrected. (Page 3, line 19)

–Line 21: "2.2 Data set A (1899-2008)" where is 2008 in Table 1? A: The table was corrected (see previous answers to remarks to page 4, lines 12 and 25).

–Line 24: "The sheet for 1916 is shown in Fig. 2" (not Fig. 3—see Line 28 comment above, may want to switch Figures 2 and 3) A: See previous answer to remark to page 2, line 28.

–Line 25: "1909-1914, and 1918-1919 are missing" A: Corrected. (Page 3, line 28)

–Page 4

–Line 7: "failure;" change comma to semicolon A: Corrected. (Page 4, line 11)

–Line 16: What are 1295 and 299 indicating? A: They are the numbers of data used for comparisons. In order to avoid confusion, we rephrased the text in brackets as: "using 1295 'surface' observations in 1964-1968" and "using 299 observations at 2-m depth in 1983-1986". (Page 4, line 20)

–Line 23: "but this was not always the case" A: The sentence was rephrased as suggested (Page 4, line 28)

–Line 24: "These analogue records were reported to 0.1 C precision, and occasionally to 0.05 C" A: Corrected. (Page 4, line 29)

–Line 33: "reported to 0.01 C precision." (remove "the") A: Corrected. (Page 5, line 6)

3. The time series at 2-m depth –Page 6

–Line 1: "the daily means are possibly estimated" "possibly" is awkwardly worded, please change A: The text was modified as follows: ". . . the daily means are estimated from data at other depths when possible.". (Page 6, line 10) –Line 9: "These correc-

tions concern estimates" "concern" is awkward, perhaps use "affect"? A: We replaced "concern" with "are associated to". (Page 6, lines 19-20)

–Line 17: "computed from those on the observation" is awkwardly worded, please change. Also, why do you need "respectively" here? A: The text was rephrased as follows: "The error on T, namely $\sigma$, is computed from the errors on the observation, $\sigma o$, and on the climatologies, $\sigma 24c$ and $\sigma c$. These errors are assessed semi-empirically as follows." (Page 6, lines 27-28)

–Line 19: "An observation is affected by" (delete "basically") A: Corrected. (Page 6, line 29)

–Line 21: "the curve thickness determine a reading error" (delete "probably") A: Corrected. (Page 7, line 1)

–Line 23: "due to turbulence and circulation. Therefore..." A: Corrected. (Page 7, line 3)

–Line 24: "representative of the hourly value" (delete "really") A: Corrected. (Page 7, line 4)

–Line 27: change to "only nominal times, and not the actual observation times, are reported." A: Corrected. (Page 7, line 7)

–Line 28: "This time uncertainty affects temperature..." A: Corrected. (Page 7, line 8)

–Line 29: "the time uncertainty until 1917" should this be 1919 or 1934 based on section 2.2? A: As stated at page 3, line 25 (original manuscript) data for 1918-1919 are missing. Uncertainty is negligible until 1917 and sometimes non-negligible from 1920 onwards. The text seems clear enough and it was not modified.

–Page 7

–Line 1: "half an hour; therefore" (change comma to semicolon) A: Corrected. (Page 7, line 12)

–Line 15: "by comparing the data sets" A: Corrected. (Page 7, line 26)

–Line 23: "as discussed in the above paragraph"? A: No, it is discussed a few lines above. In order to avoid confusion, we modified the sentence as follows: ". . . increased by 0.5 °C on the basis of the above-mentioned temperature difference." (Page 8, lines 4-5)

4. Data availability –Line 25: "as a percentage of valid days" also, valid is awkward, maybe use "total days"? A: The text was modified as follows: "Figure 4 illustrates the monthly data availability as the percentage of the number of days with temperature estimate to the number of days per month." (Page 8, lines 9-11)

–Line 26: what is meant by 80% valid days here? Same question for "valid months"? A: Although "valid" is often used to indicate that the day or month is non-missing, we rephrased the relevant sentence as follows: "Mean annual temperatures were computed for complete years, i.e. having no missing months; a missing month has less than 80% available days (Fig. 5)." (Page 8, lines 11-12) –Line 27: better word for arbitrarily? Seems very unofficial. Also, remove the repeat "Figure 4 illustrates. . ." sentence. A: We do not think that another word would be better. To the authors' knowledge no official or standard percentages are generally adopted to define a monthly mean estimate reliable. It depends on the dominant time scales of variability of the specific variable. We chose 80% because it is a high percentage (corresponding to less than one week missing in a month) although not necessarily optimal. As an example, to compute reliable monthly mean sea levels, the Permanent Service for Mean Sea Level adopts 50%. The repetition was removed.

–Page 8 –Line 2: "in the first part of the period"? A: We used "of the time series". (Page 8, line 16)

5. Conclusive remarks –Line 8: "Near-surface sea temperature data" (remove the "s" in temperatures) A: Corrected. (Page 8, line 22)

–Line 9: "from 1899 to 2008, while the second consists. . ." A: Corrected. (Page 8, line 23)

–Lines 12-13: "study sea-temperature variability on the synoptic time scale connected to meteorological forcing, and on decadal and secular time scales related to climate changes." Also, what is meant by "secular time scales"? A: There are other time scales between the synoptic and decadal ones as, for instance, those connected to atmospheric planetary waves (10-100 days), the seasonal cycle (1 year), and those connected to the variability of large-scale atmospheric patterns like ENSO (few years). We modified the sentence as follows: ". . . study sea-temperature variability on time scales from the synoptic, connected to the meteorological forcing, to decadal and century-timescales related to climate changes". (Page 8, lines 26-27)

–Line 14: "The search for undiscovered data will continue, in order to possibly fill the existing gaps." A: Corrected. (Page 8, line 28)

Author contribution –Line 16: "the data sets and led" A: Corrected. (Page 9, line 2)

–Line 17: "and collaborated on the paper writing" A: Corrected. (Page 9, line 3)

Figures

Figures 2 and 3: these are somewhat difficult to read, may want to ensure very high quality images of these are included so readers can clearly read them. A: We checked that the original 300-dpi pdf images can be magnified at least to 3 times the original size without loss of definition.

Figure 5: Can you add the 1946-2015 trend line on this figure? Also, should the caption read "from 1899 to 2015"? A: Corrected accordingly. (Page 14)

Tables Table 1: several comments as indicated above A: The table was redesigned. (Page 15)

Table 2: –Line 5: "number of observations". "instrumental error" and –What are the 1st

and 2nd numbers before and after the semicolons, e.g. 0.18; 0.31? Finally, can you somehow indicate that the errors are $\frac{1}{2}$ the band (e.g. +/- 0.15 C)? A: The table was redesigned and the caption rewritten. Concerning the errors, they are always $\frac{1}{2}$ the band (see sect. 3.2), therefore there should not be any confusion. (Pages 15-16)

Please also note the supplement to this comment:
https://www.earth-syst-sci-data-discuss.net/essd-2019-15/essd-2019-15-AC1-supplement.zip

---

## Author Comment (AC2) · 8 May 2019

Pages and lines of the edited manuscript, unless otherwise stated (To make reading easier, the pdf version of this document and of the edited article, with revisions, are attached as supplement2.zip.)

It seems that the paper could gain in relevance if two points would be added. I. There is no comparison with the trend of (surface) sea temperatures of either coastal, nor 'global' ocean sea surface temperature data. This 'global warming trend' is a hot topic, relevant nowadays. Authors confined themselves mostly to the methodology of 'combining' the data of different measurements techniques, of different sea temperature 'sampling', on elaborating the time series (filtering the data) and on the trend of sea temperature rise that they reveal from those data. There are certainly many research papers that describe centennially temperature trends elsewhere. Moreover, there are reports of IPCC (although quality reports are lately blurred with reports of IPCC meetings. . .) that still somehow 'matter', e.g. the IPCC Report 'Global warming of 1.5°C', in Chapter 1: [Allen, M.R., et al., 2018: Framing and Context. In: Global Warming of 1.5°C. An IPCC Special Report on the impacts of global warming of 1.5°C above pre-industrial levels and related global greenhouse gas emission pathways, in the context of strengthening the global response to the threat of climate change, sustainable development, and efforts to eradicate poverty [Masson-Delmotte, et al. (eds.)]. In Press.]. There one may find a few 'useful sentences' already at the beginning, e.g.: 'Human-induced warming reached approximately 1°C (likely between 0.8°C and 1.2°C) above pre-industrial levels in 2017, increasing at 0.2°C (likely between 0.1°C and 0.3°C) per decade (high confidence)', and also 'Accordingly, warming from preindustrial levels to the decade 2006–2015 is assessed to be 0.87°C 2 (likely between 0.75°C and 0.99°C).' These sentences are just very modest examples about how the result (the temperature trend in the 'intestines' of the central middle Europe, facing the sea) of authors makes sense and is 'in line' with the trends others have found. There are also differences (e.g. in the trend within last 30 years) with other findings, which would well be described in Discussion. In the Introduction, though, the relevance of this particular, long time series has to be emphasized and compared with other very long term studies. II. The second topic for which it seems just to be linked to the paper, is the matter of the sea-level rise. A brief look on publications of authors clearly shows that at least one of them has a solid reputation in 'knowing this matter well'. Authors may relatively easily combine their sea temperature rise finding with the sea level rise simply due to steric effect – they can estimate it and may also estimate the error of the estimation (they showed how nicely they know how to estimate errors. . .) of sea level rise due to temperature expansion of water (e.g. the effect of salinity (variability)). There is quite a large number of papers over the Adriatic and the Mediterranean

Sea that handle separately the sea level rise and the temperature rise, but only a few link these two trends. This is a good chance 'to do it right'! Answer (to both I and II): We did not extend the paper because in the journal's website (www.earth-system-science-data.net/about/aims_and_scope.html) it is written that "Any interpretation of data is outside the scope of regular articles.". The comparison of the trends obtained for different locations and from the global ocean requires data interpretation. The connection of sea-temperature rise with sea-level rise is a subject deserving a paper on its own. That is why we did not include in the article anything but the data description and the time series homogenization. We think that the text should not be extended the include the reviewer's suggestions.

Specific comments

Page, 1. Line 16: is the text in this line in 'bold'? A: This question is unclear. However, from the pdf version it does not seem so.

Page 3, line 24: Fig. 3 is referred. Should it be the Fig. 2? There was no Fig. 2 before in the text and it looks from Figure and figure caption of Fig. 2 that this should be Fig. 2. A: We are sorry for the mistake. Figures 2 and 3 at the end of the manuscript were swapped and identified by the wrong number. The mistake was corrected. (Pages 12-13)

Page 5, line 13: $T0(h,d,m,y,z) \rightarrow T0(h,d,m,y,z)$ A: This remark is unclear.

Page 5, line 18: '...between 13 and 17 values of T0.' Could it be added 'out of (?) 24 × 365.25' on average per year? A: In equation 1 it is clearly written that $Tc(h,d,m,z)$ is the ratio of two sums over y from 1999 to 2015, i.e. 17 elements, while '24 x 365.25' is the average number of hours per year, which is not involved. The text was not modified as it seems clear enough.

Page 5, expression (3): In the expression (2) T24c is written down. However, it somehow follows from the expression (3) and the comment below it that T24c should be

expressed as the average of N values od Tc the number of available observations on the relevant day, and not the average of '24' values (expression (2)). Correct? A: No. The text at page 5, lines 11-12 reads "obtained by averaging hourly (0-23) temperatures and mean daily temperatures of days when all the 24 hourly observations were available". Therefore, 24 values are always available for the average.

Page 6, expression (5): It looks OK. . . A: This remark is very unclear.

Page 7, line 9: 'observational error $\sigma 0$=0.18 °C, we obtain $\sigma c$=0.05 °C and $\sigma 24 c$=0.01 °C' →ïĂăobservational error $\sigma 0$=0.18 °C, we obtain $\sigma c$=0.05 °C and $\sigma 24 c$=0.01 °C. A: If we understand it correctly, the reviewer suggests to remove italics for numbers. It has been corrected. (Page 7, line 20)

Page 7, line 23: '. . .was increased by 0.5 °C, as discussed above.'. Do authors refer to the line 18 in which $\Delta T = 0.5 \pm 0.5$ °C is written? If so, then they could write this more clearly and on line 18: $\Delta T = 0.5 \pm 0.5$ °C →ïĂăïČĎT = $0.5 \pm 0.5$ °C. The same for another $\Delta$Tin the same line. A: In order to avoid confusion, we modified the sentence as follows: ". . . increased by 0.5 °C on the basis of the above-mentioned temperature difference." Also in this case we removed italics from numbers. (Page 7, lines 29-30)

Page 7, line 27: there is a redundant copy of the sentence about Figure 4 from the line 25. . . . A: The repeated text was removed. (Page 8, line 12)

Please also note the supplement to this comment:
https://www.earth-syst-sci-data-discuss.net/essd-2019-15/essd-2019-15-AC2-supplement.zip